# Ash Presence and Abundance Derived from Composite Landsat and Sentinel-2 Time Series and Lidar Surface Models in Minnesota, USA

**Trevor K. Host [1], Matthew B. Russell [1],\***, **Marcella A. Windmuller-Campione [1], Robert A. Slesak [1,2] and Joseph F. Knight [1]**

[1] Department of Forest Resources, University of Minnesota, 1530 Cleveland Ave. N., St. Paul, MN 55108, USA; hostx009@umn.edu (T.K.H.); mwind@umn.edu (M.A.W.-C.); raslesak@umn.edu (R.A.S.); jknight@umn.edu (J.F.K.)

[2] Minnesota Forest Resources Council, 1530 Cleveland Ave. N., St. Paul, MN 55108, USA

\* Correspondence: russellm@umn.edu

**Abstract:** Ash trees (*Fraxinus* spp.) are a prominent species in Minnesota forests, with an estimated 1.1 billion trees in the state, totaling approximately 8% of all trees. Ash trees are threatened by the invasive emerald ash borer (*Agrilus planipennis* Fairmaire), which typically results in close to 100% tree mortality within one to five years of infestation. A detailed, wall-to-wall map of ash presence is highly desirable for forest management and monitoring applications. We used Google Earth Engine to compile Landsat time series analysis, which provided unique information on phenologic patterns across the landscape to identify ash species. Topographic position information derived from lidar was added to improve spatial maps of ash abundance. These input data were combined to produce a classification map and identify the abundance of ash forests that exist in the state of Minnesota. Overall, 12,524 km$^2$ of forestland was predicted to have greater than 10% probability of ash species present. The overall accuracy of the composite ash presence/absence map was 64% for all ash species and 72% for black ash, and classification accuracy increased with the length of the time series. Average height derived from lidar was the best model predictor for ash basal area ($R^2 = 0.40$), which, on average, was estimated as 16.1 m$^2$ ha$^{-1}$. Information produced from this map will be useful for natural resource managers and planners in developing forest management strategies which account for the spatial distribution of ash on the landscape. The approach used in this analysis is easily transferable and broadly scalable to other regions threatened with forest health problems such as invasive insects.

**Keywords:** Google Earth Engine; Landsat; lidar; *Fraxinus*; forest health

---

## 1. Introduction

Remote sensing allows for the broad-scale monitoring of forests around the globe using precise spatial information and frequently repeating observations to identify and detect changes in vegetation presence, abundance, and condition. These datasets lead to greater knowledge of forest ecosystem patterns over large spatial extents [1]. Remote sensing has a long history in aiding forest inventory, including the identification of forest cover in the form of geospatial maps, assessment of forest health as related to defoliation and discoloration [2], and quantitative models that provide predictions such as timber volume, basal area, aboveground biomass, and carbon [3]. Derived maps and reports are used to enhance on-the-ground measurements, identify areas of interest for further investigation, and provide consistent, wall-to-wall data coverage to inform land management decisions [4,5].

The ash genus (*Fraxinus* spp.) is widely distributed across much of central and northern North America both in urban and traditional forest environments. Since the identification of the invasive

emerald ash borer (*Agrilus planipennis* Fairmaire; EAB) in 2002 in Detroit, Michigan, USA, there has been widespread mortality of ash across North America [6]. Currently, EAB is present in 35 U.S. states and several Canadian provinces. In Minnesota, USA, EAB is locally present in several counties in the central and southern regions of the state. The EAB larvae feed on the phloem of ash trees and contribute to tree mortality within one to five years of infestation. After introduction of EAB, annual ash mortality can increase by as much as a 2.7% in a county [7]. The anthropogenic movement of the beetle through infested firewood or other products has aided in the introduction of EAB to new, distant locations [6].

Ash is a prominent tree species in Minnesota forests that exists across many land cover types. Miles et al. [8] estimated that there are 1.1 billion ash trees in Minnesota that are at least 2.5 cm in diameter at breast height (DBH) or greater, accounting for 8% of all trees in the state. However, due to its low economic value, there has been limited research on this forest type, including basic forest inventory to understand the extent and locations of ash stands. The three species of ash found in Minnesota are black (*Fraxinus nigra* Marshall), green (*Fraxinus pennsylvanica* Marshall), and white ash (*Fraxinus americana* L.). The majority of the individuals are black and green ash with relatively few white ash trees. In Minnesota, black ash is dominant in two vegetation types: (1) Black ash-elm/trillium vegetation communities that occupy moist sites with deep organic soils and (2) black ash/yellow marsh marigold vegetation communities on sites with better drainage. Black ash are dominant in the overstory of swamp forests that occur in extensive complexes, in topographically low depressional areas, and at transitions between upland forests and peatlands [9]. However, black ash can be found in lower abundance in nearly all vegetation types where it mixes with other species [10].

Historical disturbances have recently been mapped using the full extent of the Landsat archive across Minnesota's forests, highlighting the value imagery time series to monitor forest dynamics [11]. Remote sensing at the individual species level is difficult in mixed-species forests like Minnesota's because of the numerous species and high degree of heterogeneity in small areas, i.e., the size of stands where forest management decisions are primarily made. Fortunately, ash has certain biological and physiographic features that may help to differentiate it from other tree species. Deciduous phenology patterns are useful for speciation at a broad level (e.g., deciduous vs evergreen). Ash species drop their leaves earlier than other deciduous species in the fall [12]. This is useful from a remote sensing perspective because the change in optical reflectance from healthy, green leaves to barren, grey branches can be an indicator of species. However, differences in local site conditions and climate influence phenology as well, and additional information is likely required to accurately identify species. A physiographic characteristic of ash, particularly in black ash, is the prevalence of the species in relatively low elevation positions with high degrees of soil moisture such as swamps, bogs, fens, and other forested wetlands. These characteristics separate ash from other deciduous tree species like aspen (*Populus* spp.), one of the most common hardwood trees in Minnesota.

Given the extent of ash and the impending EAB threat in Minnesota, there is a need for accurate and high-resolution maps of ash presence and abundance. These maps will help to quantify the current extent of ash for future analysis and develop land management plans for areas of high EAB infestation risk. Previous maps of ash abundance have been generated at coarse spatial resolutions, i.e., 250 m [10], using alternative imputation methods [13], or within single Landsat scenes [14], but not at a moderate satellite spatial resolution for the regional extent. The objective of this work was to produce a 30-m resolution map of current ash presence/absence in Minnesota. Ash abundance was modeled in terms of basal area using lidar height metrics.

## 2. Materials and Methods

### 2.1. Study Area and Imagery

#### 2.1.1. Study Area

The area of study was the state of Minnesota, USA, which includes four ecological provinces: The Laurentian Mixed Forest, Eastern Broadleaf Forest, Prairie Parkland, and Tallgrass Aspen Parklands [15]. Minnesota's land cover is diverse, with approximately 50% of forested cover [16]. Minnesota's forest ownership is also diverse, with 55% in public ownership, consisting of lands controlled by the state (23%), federal (17%), and county and local governments (15%) [8]. There is a strong gradient from agricultural cover in the southwest to primarily forested cover in the northeast. A wide range of forest types occur across Minnesota's 6.4 million hectares of timberland, with the most common types being aspen-birch and spruce-fir. Yet, ash forests are a major cover type, with approximately 8% of tree species in the state [8]. Because of the extent of ash and the current localized infestation of EAB, Minnesota represents an opportunity to explore the use of remotely sensed data to provide valuable information on extent of the ash resource to forest managers.

#### 2.1.2. Optical Imagery

The sources of optical imagery included Landsat 5 TM (1984–2011), Landsat 7 ETM+ (1999–present), Landsat 8 OLI (2013–present), and Sentinel-2 MSI (2015–present). Each collection was filtered to scenes overlapping the Minnesota state boundary. The Landsat collections were processed to a level-one terrain corrected (L1T) product [17]. Gaps in Landsat 7 imagery caused by the 2003 Scan Line Corrector error were masked using a per-pixel bit mask and not included in further analysis. L1T surface reflectance data products are terrain-corrected and radiometrically calibrated using the Landsat Ecosystem Disturbance Adaptive Processing System (LEDAPS) (TM, ETM+) [18] and Landsat Surface Reflectance Code (LaSRC) (OLI) [19], which includes per-pixel cloud masking using CFMASK [20]. The Normalized Difference Vegetation Index (NDVI) was used to identify forested areas and calculated for each sensor using the NIR and Red bands:

$$\text{NDVI} = \frac{NIR - Red}{NIR + Red} \tag{1}$$

Forested areas were identified using a multi-threshold filter where the NDVI value was greater than or equal to 0 and the lidar canopy height was greater than or equal to 3.

#### 2.1.3. Google Earth Engine (Time Series)

The Google Earth Engine (GEE) platform was used for processing Landsat imagery and producing the classification product. GEE provides ready access to the archive of L1T surface reflectance Landsat data through a browser-based programming interface. All additional layers were uploaded through user account storage as image assets. The GEE platform was used in a number of ways. First, GEE filtered out clouds and poor-quality pixels from each satellite imagery collection. Second, it calculated NDVI for each image in the collection. Last, it calculated a harmonic regression of NDVI through time and perform the thematic classification of pixel values using a RandomForest algorithm.

Harmonic regression of Landsat NDVI observations have been demonstrated by Brooks et al. [21]. A fixed harmonic regression of per-pixel NDVI observations was fit to each cloud-masked imagery archive. The frequency of the harmonic was fixed to one period per year to match the seasonal vegetation pattern. Each time series was approximated as a trigonometric polynomial, where $t$ is the image timestamp, $\omega$ is the harmonic period, $e$ is the error in the model fit.

$$\text{NDVI}_t = \beta_0 + \beta_1 t + \beta_2 \cos(2\pi\omega t) + \beta_3 \sin(2\pi\omega t) + e \tag{2}$$

The root mean squares of the residual errors (RMSE) were output as a band for evaluation. The estimated coefficients were used to compute the annual amplitude and phase of the single cycle harmonic regression using the following equations from Shumway and Stoffer [22]:

$$Amplitude = \sqrt{{\beta_2}^2 + {\beta_3}^2} \tag{3}$$

$$phase = atan(\beta_3 / \beta_2) \tag{4}$$

The annual NDVI amplitude and phase computed bands were used as inputs to the classification procedure. For visualization, we organized time series by Day-Of-Year (DOY) of image acquisition, where DOY 1 represents January 1 and DOY 365 represents December 31 [23].

### 2.1.4. Lidar

Aerial lidar data were acquired for the state of Minnesota between 2008 and 2011. The lidar point cloud had a minimum point density of 1.5 pulses/m$^2$. The vertical accuracy RMSE$_z$ was less than 15 cm. The software package LAStools was used to create several interpolated models describing Canopy Height Model (CHM) and Digital Elevation Model (DEM) at 3-m spatial resolution. From the DEM, Topographic Position Index (TPI) and Compound Topographic Index (CTI) were created at a spatial resolution of 3 m for the state. The TPI raster is an index that represents the local difference in elevation, measured by the elevation difference between each cell in a DEM to the mean elevation of the neighborhood of surrounding cells. The CTI raster is an elevation index that incorporates topographic position with flow accumulation from upstream drainage areas [24]. The (CTI) is defined as ln[(α)/(tan (β)], where α represents the local upslope contributing area and β represents the local slope gradient. The topographic variables were resampled to 30-m resolution using the mean value prior to modeling.

### 2.2. Model Development

### 2.2.1. Forest Inventory Data

A RandomForest [25] classifier was developed with a statewide forest inventory database as training data. RandomForest is a machine learning approach to predict the state of a variable, in our case, the presence or absence of ash trees, based on a list of input variables. The input variables are split into a sequence of steps or a decision tree that best separates the training data. RandomForest creates many decision trees and combines the results to provide a model with best prediction results. The parameters for RandomForest include the user defined number of decision trees and (m) number of variables per split. When building a set of regression trees, 20% of samples are withheld from the RandomForest algorithm (the out-of-bag samples) and are evaluated in determining the model's accuracy [25]. The Minnesota Department of Natural Resources maintains a continuously updated polygon database used for forest management applications, termed the Forest Inventory Management (FIM) database [26], which was used in this analysis (Figure 1).

The FIM dataset contained detailed attributes of spatial stand boundaries that were digitized from aerial photography. The FIM dataset was filtered in several ways to ensure only the most homogeneous and up-to-date stands were selected as training data. As an example, only stands measured between 2000 and 2018 were included in this analysis. Stands where ash species were identified as the main cover type and primary species in the stand were selected (Figure 1). The FIM dataset was sampled at a 30-m resolution, effectively converting polygons to raster to match the input variables. This resulted in 2498 samples for training. A random selection of samples were chosen using the sampleRegions command in GEE to train the classifier at a scale of 30 m on six predictor variables including NDVI amplitude, NDVI phase, NDVI median, canopy height model, lidar TPI, and lidar CTI (Table 1). Once the model was fit to the training data, the model was used to infer the probability of ash classification for each pixel in the input imagery. After filtering the FIM polygons by survey year, dominant species, and forested land type, there were 24,467 polygons and 498 ash-dominant polygons where ash was the

main cover type and primary species in the stand (approximately 2% of the data). A concept flowchart of the classification process is shown in Figure 2.

The NDVI time series of the three Landsat and one Sentinel-2 sensor was created for input to RandomForest classification of ash forests (Table 1). The RandomForest model parameters were set to 10 decision trees with two input variables per split (m). The input variables per split was determined by the standard square root of predictor variables. The annual amplitude, phase, and median of NDVI was calculated for each 30-m pixel. The number of NDVI observations varied by pixel depending on duration of satellite operation and image swath overlap. The RMSE of the NDVI harmonic model represents the error in predicted values of NDVI amplitude, phase, and median. These intermediate datasets were developed to be incorporated as primary inputs to the RandomForest classification model.

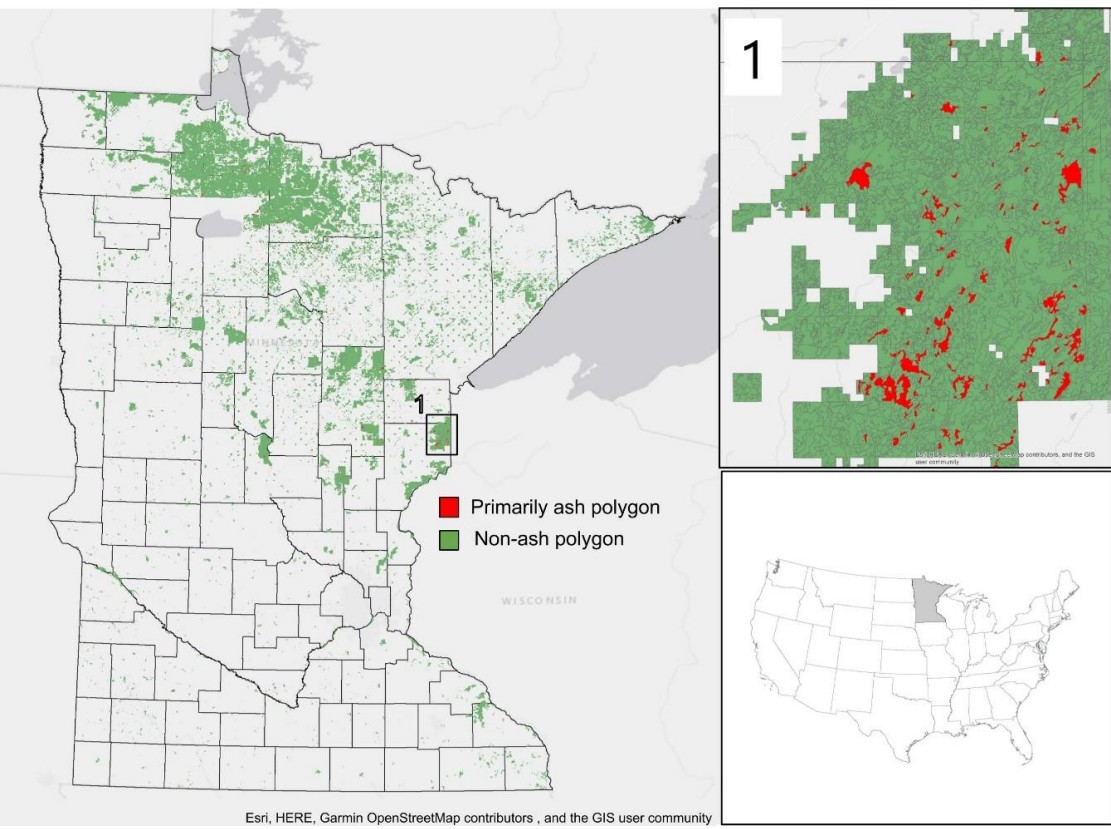

**Figure 1.** Illustration of the presence and absence of ash stands based on the Minnesota's Forest Inventory Management polygon database. Inset displays an example of diverse Minnesota forests that transition from non-ash to ash-dominated.

**Table 1.** RandomForest data inputs for classifying ash presence and absence in Minnesota, USA.

| Predictor | Resolution (m) | Abbreviation | Description | Source |
|---|---|---|---|---|
| *NDVI Amplitude* | 30 | $NDVI_a$ | Magnitude of NDVI difference 1 yr cycle | TM, ETM+, OLI, MSI |
| *NDVI Phase* | 30 | $NDVI_p$ | Day of year position of maximum NDVI | TM, ETM+, OLI, MSI |
| *NDVI Median* | 30 | $NDVI_m$ | Median of NDVI observations | TM, ETM+, OLI, MSI |
| *Canopy Height Model* | 3 | CHM | Max height of returns above ground level | ALS |
| *Topographic Position Index* | 3 | TPI | Local difference in elevation | ALS |
| *Compound Topographic Index* | 3 | CTI | Soil moisture potential calculated from a DEM | ALS |

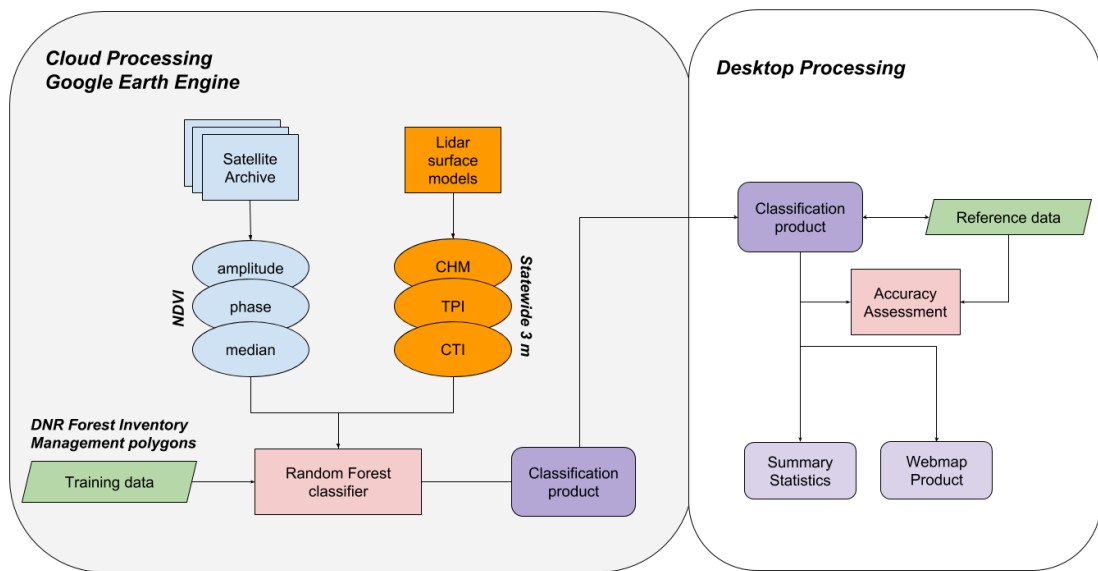

**Figure 2.** Concept flowchart of model development for ash classification. The Google Earth Engine platform was used for processing Landsat surface reflectance imagery and producing the classification product. Server-side computing enabled use of the Landsat Archive and allowed easy modification to the analysis. Validation was performed outside of the cloud environment to ensure data security and to allow integration with additional GIS layers.

Clouds and poor quality pixels were masked from the Sentinel-2 top-of-atmosphere imagery. The NDVI was used to normalize the reflectance values to be comparable. Sentinel-2 imagery was processed at 10-m resolution but resampled to 30-m resolution using the mean value resampling when combined with Landsat imagery. To determine ash presence, a 10% probability of occurrence was set because this was the rate at which errors of commission and omission of ash presence were minimized.

### 2.2.2. Basal Area Modeling

The abundance of ash in terms of basal area was modeled with metrics derived from the lidar canopy height model (CHM) at 3-m resolution. A number of forest inventory plots from county and state lands, as well as detailed measurements from ongoing black ash research [27], were compiled to train the basal area prediction model (Figure 3; Table 2). The datasets included forest inventory plots where species and diameter at breast height (DBH) were consistently measured, but with some differences in plot size. The Chippewa-DNR inventory (CDI) was a systematic sample of 147 fixed-radius plots that were 404 m$^2$ in size, in which trees with DBH greater than 8.9 cm were recorded (Figure 3(1)). The Cloquet Forestry Center Inventory (CFI) was a sample of 349 fixed-radius plots in which trees greater than 12.7 cm in DBH were measured on plots 578 m$^2$ in size (Figure 3(2)) [28].

In total, 618 field measured plots were used. Basal area was calculated for each inventory plot to estimate total plot basal area. From the lidar derived CHM, several metrics were computed: Average height (CHM$_{avg}$), standard deviation of height (CHM$_{sd}$), minimum height (CHM$_{min}$), maximum height (CHM$_{max}$), range of height (CHM$_{rng}$). The UTM northing and easting and plot area were recorded in the compiled data but were not used in the model. Variables were added stepwise to the multivariate regression model using R software [29]. Once the model was fit to the extracted lidar metrics and if ash was present, the prediction model was applied to produce an aggregated 30-m raster map of ash basal area concurrent with the ash prediction map.

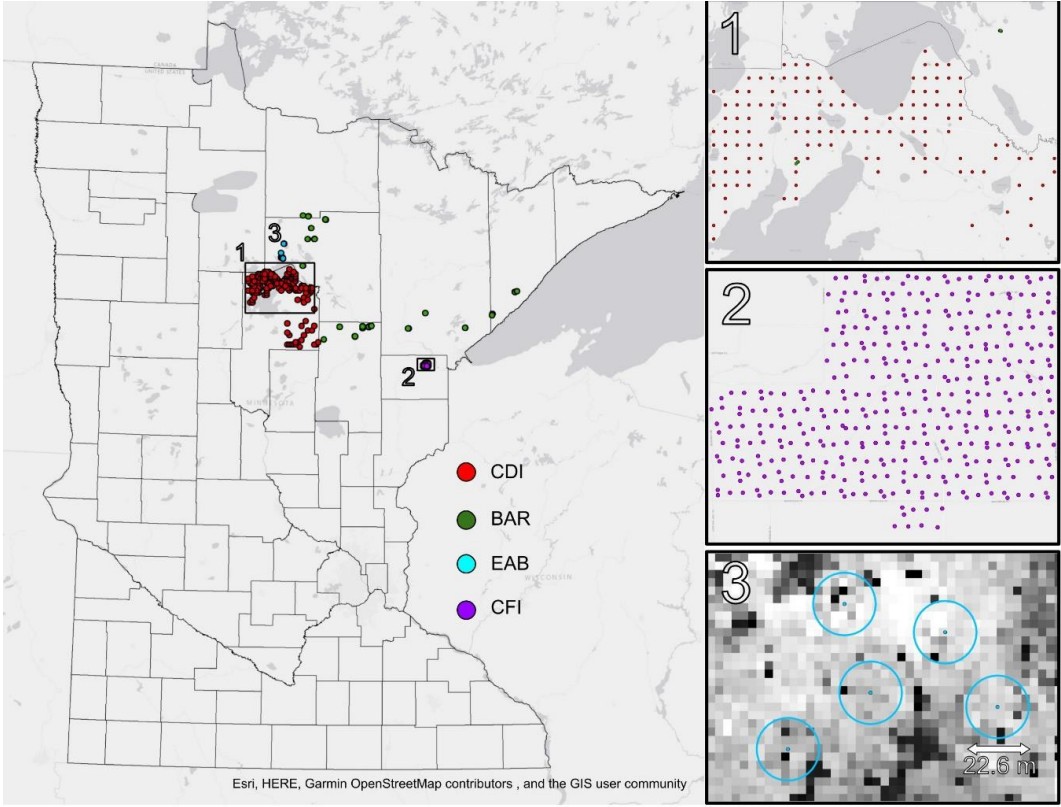

**Figure 3.** Location of forest inventory plots with county boundaries from a variety of county, state, and research plots used to train a spatial predictive model of ash basal area in Minnesota, USA, including inventory plots from Chippewa-Department of Natural Resources (CDI), black ash research (BAR), emerald ash borer research (EAB), and Cloquet Forestry Center plots (CFI). Insets show (**1**) CDI systematic sample plots, (**2**) CFI inventory plots, (**3**) raster resolution view of the CDI inventory plots.

**Table 2.** Forest inventory plot measurements used to train a spatial predictive model of ash basal area based on imputation of lidar derived canopy height model metrics in Minnesota, USA.

| Dataset | Abbr | Plot Size (m$^2$) | Year | Total n | Mean | SD | Min | Max |
|---|---|---|---|---|---|---|---|---|
| | | | | | Ash basal area (m$^2$/ha) | | | |
| Black ash research plots | BAR | 404 | 2018 | 80 | 18.62 | 6.63 | 0.00 | 41.21 |
| Chippewa-DNR inventory | CDI | 404 | 2017 | 147 | 23.44 | 14.55 | 0.36 | 68.81 |
| EAB research plots | EAB | 404 | 2015 | 42 | 28.92 | 6.53 | 9.54 | 48.71 |
| Cloquet forest inventory | CFI | 578 | 2014 | 349 | 23.93 | 12.81 | 0.25 | 72.42 |

*2.3. Model Validation*

Forest Inventory and Analysis

We used USDA Forest Service data from the Forest Inventory and Analysis (FIA) program to independently assess model performance for both presence/absence and abundance. Each plot was comprised of four subplots, with one central and three peripheral subplots located 36.58 m from the central subplot at azimuths of 0, 120, and 240 degrees. Subplots were 7.32-m fixed radius, where trees 12.7 cm DBH and larger were measured. Plots from the five-year evaluation period between 2013 and 2017 (5924 total) were used as independent reference data to validate the predicted ash presence/absence classification product (Figure 4). Of those plots, 1763 contained at least one live ash tree. The true coordinates of FIA plot locations were used to extract the classification value of the nine pixels coincident with the plot. However, to maintain privacy and the integrity of the plots, the publicly available locations of the plots are presented in tables and figures. To assess the accuracy of

the classified map, the relative proportion of ash:total basal area was incorporated into validation point selection. The sensitivity of basal area and probability of classification were assessed. The accuracy of detection of each individual species was also reported.

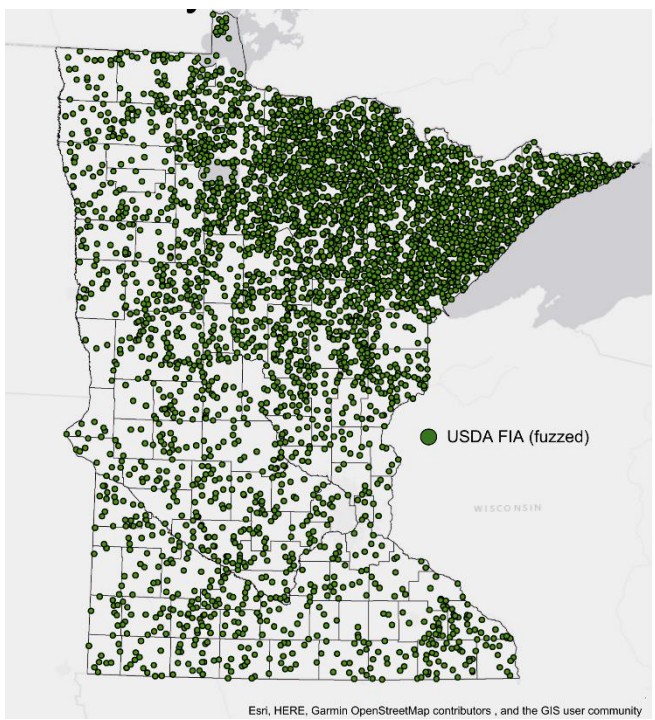

**Figure 4.** Location of 5924 permanent Forest Inventory and Analysis plot locations (fuzzed) with county boundaries in Minnesota, USA from 2013 to 2017 used in model validation and classification accuracy assessment.

## 3. Results

### 3.1. Ash Mapping

#### 3.1.1. Ash Presence/Absence

The predicted probability of ash presence/absence produced a 30-m resolution probability raster. The probability distribution was right-skewed, where high probability was relatively more rare and low probability predictions were more common. A threshold value of 0.10 was established to define the most appropriate presence probability based on the classification sensitivity. The most important variables to the classification accuracy were the $NDVI_a$ and $NDVI_m$. The accuracy varied more by sensor origin (Table 3).

**Table 3.** Out-of-bag accuracy to assess the success of model classification with differing input variables using resubstitution of training data (TPI = Topographic Position Index, CTI = Compound Topographic Index).

| Sensor | Time Series | Time Series, TPI | Time Series, CTI | Time Series, TPI, CTI |
|---|---|---|---|---|
| Landsat 5 | 80% | 82% | 82% | 84% |
| Landsat 7 | 80% | 83% | 83% | 84% |
| Landsat 8 | 75% | 77% | 78% | 79% |
| Sentinel-2 | 78% | 81% | 80% | 81% |

The results of ash detection from each sensor time series ranged between 29% and 46% and the composite of all sensors was the highest with 64% detection. The accuracy tended to increase with the length of the time series (Table 4).

**Table 4.** NDVI time series accuracy summary table by satellite sensor.

| Time Series | Acquisition Years | Ash Detected | Ash Undetected | Total Ash Plots | Detection Accuracy |
|---|---|---|---|---|---|
| Landsat 5 | 1985–2011 | 573 | 684 | 1254 | 46% |
| Landsat 7 | 1999–present | 538 | 719 | 1254 | 43% |
| Landsat 8 | 2013–present | 470 | 787 | 1254 | 37% |
| Sentinel-2 | 2015–present | 369 | 888 | 1254 | 29% |
| Composite | 1985–present | 797 | 460 | 1254 | 64% |

Overall, 1,252,400 ha (12,524 km$^2$) of forested land was predicted to have greater than 10% probability of an ash species present (Figure 5). The distribution of ash generally followed the forested land cover. The majority of the predicted ash presence was located in the north-central region of the state. Within the forested region of the state, there was lower density of ash predicted in the northeast region, which began to transition to boreal forest.

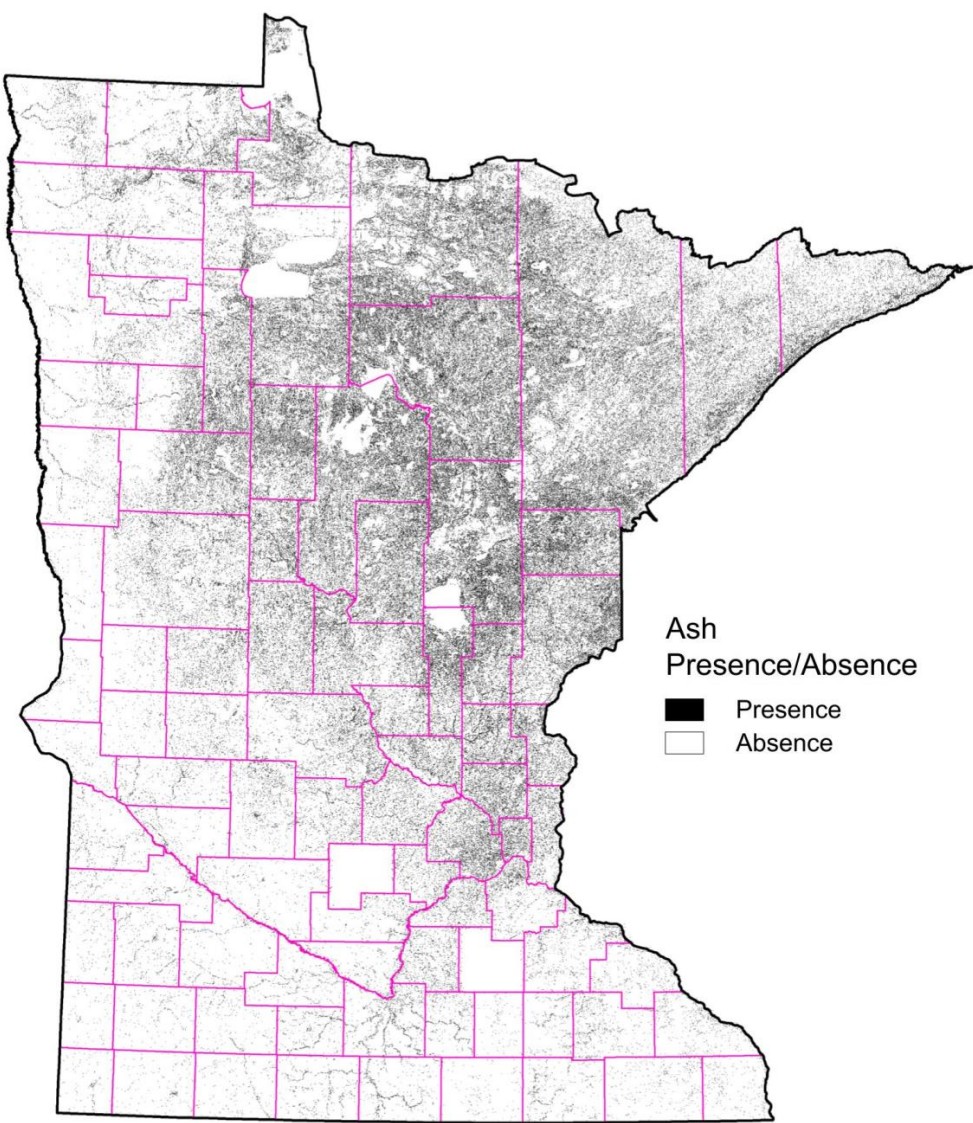

**Figure 5.** Presence/absence of ash resulting from RandomForest predictions at 30 m with county boundaries in Minnesota, USA.

### 3.1.2. Ash Abundance

Stepwise linear models were fit to a compiled dataset of 618 field observations of basal area (BA) per plot:

$$BA = \beta_0 + \beta_1 CHM_{mean} + \beta_2 CHM_{std} + \beta_3 CHM_{max} + \beta_4 CHM_{min} + \beta_5 CHM_{range} \tag{5}$$

The CHM$_{mean}$ was the most important variable in fitting the model. CHM$_{std}$, CHM$_{max}$, CHM$_{min}$, and CHM$_{range}$ were all nonsignificant when paired with CHM$_{mean}$. The basal area model fit of predicted estimates was found to have an overall $R^2 = 0.40$ but varied by source. A linear fit model from the Chippewa-DNR Inventory and Cloquet Forestry Center Inventory datasets had the highest $R^2$ values at 0.65 and 0.64, respectively.

When ash was present on a plot, the basal area of the species averaged 16.1 m$^2$ha$^{-1}$. Basal area followed a similar spatial distribution as the presence/absence classification map, with a higher density of ash abundance in the central portion of the state (Figure 6).

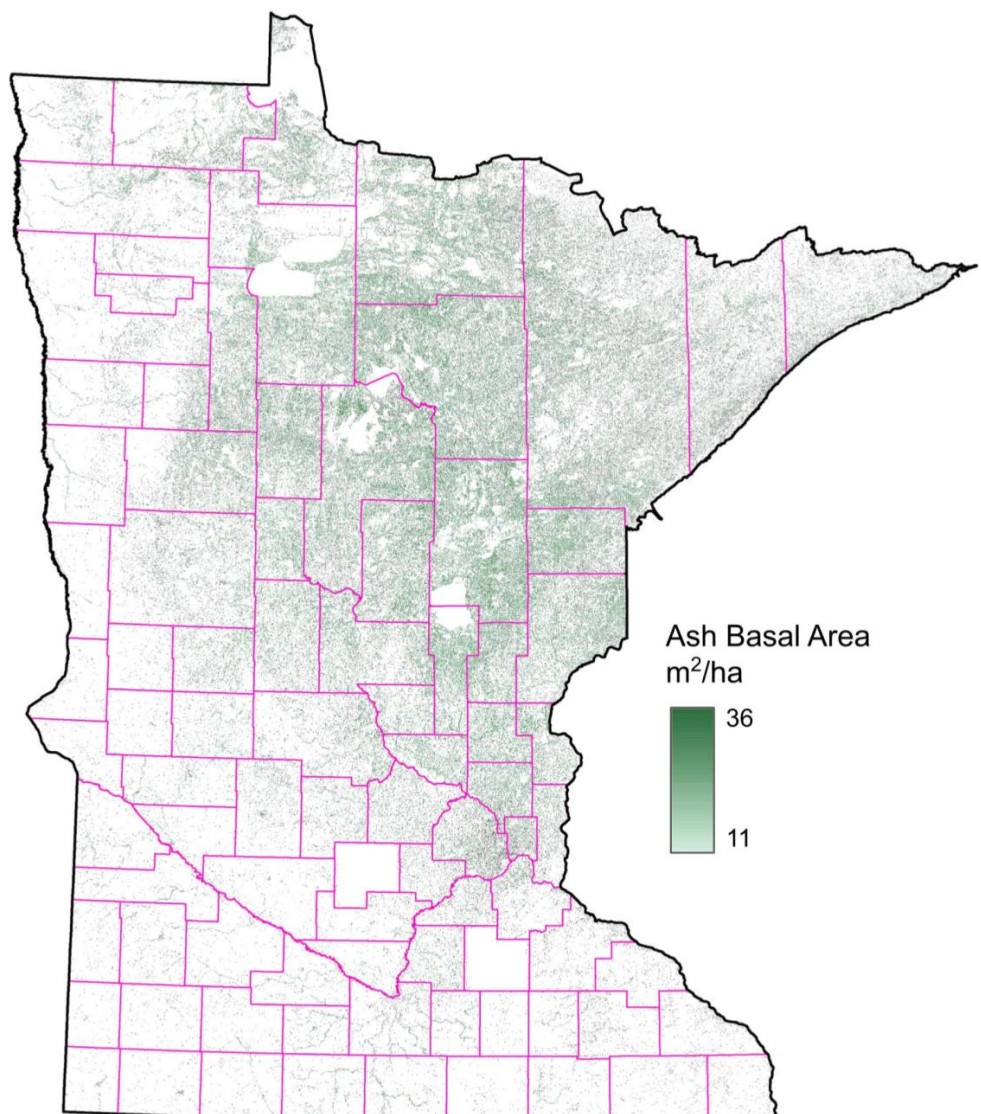

**Figure 6.** Model-derived 30-m ash abundance map depicting basal area based on presence/absence map and lidar canopy height model with county boundaries in Minnesota, USA.

*3.2. Validation*

FIA Accuracy Assessment

Of the 1761 plots that contained at least one live ash tree, we only used plots that contained ash basal area greater than or equal to 10% of total basal area on the plot for validation. There were 1254 plots where live ash composed at least 10% of the basal area that was used to evaluate the success of detection of the classification map. Plots that contained no ash trees may not represent the complete absence of ash because species were measured on only a sample of the plot and we did not confirm total absence of ash species in the encompassed area.

The overall accuracy of the composite ash presence/absence map was 64% (n = 5922) for all ash, 72% for black ash (*n* = 868), and 53% for green ash (*n* = 504) (Tables 5 and 6).

**Table 5.** Confusion matrix to assess accuracy of ash presence/absence for Forest Inventory and Analysis plots in Minnesota, USA.

| Source | Actual Presence | Actual Absence | Total | User's Accuracy |
|---|---|---|---|---|
| Predicted presence | 797 | 1687 | 2484 | 32% |
| Predicted absence | 457 | 2981 | 3238 | 87% |
| Total | 1254 | 4668 | 5922 | - |
| Producer's accuracy | 64% | 64% | - | 64% |

**Table 6.** Accuracy of ash species detection using Forest Inventory and Analysis data in Minnesota, USA.

| Source | Ash Detected | Ash Undetected | Total Ash Plots | Detection Accuracy |
|---|---|---|---|---|
| All ash | 797 | 460 | 1254 | 64% |
| Black ash | 625 | 243 | 868 | 72% |
| Green ash | 267 | 237 | 504 | 53% |

## 4. Discussion

The detection of ash species using time series at 30-m spatial resolution using composite imagery and cloud-computing software like Google Earth Engine provided a new possibility to assess ash presence. Combined with lidar data, ash abundance was determined across Minnesota in units of basal area. Distinguishing individual species from optical imagery is notoriously difficult, largely because the spatial resolution of imagery causes specific vegetation signatures to become mixed with other species [30]. This is especially true in heterogeneous environments such as the Laurentian mixed forest where ash is more abundant. Our approach using NDVI metrics to account for the unique phenology of ash helped to address this limitation and distinguish differences in vegetation. Imagery with higher spatial resolution would reduce this issue but there are currently no such time series with sufficient duration.

The utility of large datasets, such as the Landsat archive, depend on efficient computation ability. Pixel-based time series modeling using hundreds of NDVI images would not be possible using traditional geospatial processing methods. Google Earth Engine facilitated the dense time series analysis through the use of existing cloud masking and reflectance correction that would have otherwise been a barrier to model development. The spatial resolution of the imagery was not adequate to observe individuals, but the phenologic pattern derived from the time series seemed to have captured species-specific patterns. Whereas the time series from different sensors encompasses a wide range and differing timespans, the number of observations from each sensor time series seemed to have greater impact than the "currentness" of the time series. Observations of seasonal fluctuations in vegetation describes ecological patterns that added value to the predictive model in all cases.

Whereas time-series imagery is beneficial to examine forest disturbances that occur quickly at a high magnitude [11], e.g., fire or timber harvesting, and damages from insects, a relatively slow-acting

change agent is also possible to examine using Landsat data [31]. The application of these data serve a broader understanding of where ash forests exist in Minnesota and how much impact EAB will have on the state's forest resources. Our detection accuracy of black ash (72%) aligns well with the accuracy rate of 85.5% observed by the authors of [14] for the species in a single Landsat scene in northern Minnesota and the rate of 89.5% observed by the authors of [32] across the northeastern Minnesota region. Our slightly lower detection accuracy compared to these studies was likely the result of the broader geographic area examined (i.e., statewide) and the difference in dates between lidar acquisition (2008–2011) and the field inventory measurements (2014–2018). Similarly, historical Landsat data are older compared to Sentinel-2 data, which also differ in acquisition dates from field inventory measurements. Compared to previous estimates of the extent of ash in Minnesota, our value of 1.25 million hectares aligns well with the design-based estimate of ash abundance provided by the FIA program. The FIA program estimates that ash is at least 50% of the total live tree volume on 0.45 million hectares of forest land but is a component of 1.7 million additional hectares of forest land where it occurs with other tree species [33]. The distribution and abundance of ash observed in this analysis also generally agree with the statewide results of Wilson et al. [13] and Kurtz et al. [10]. Our approach was most similar to that of Engelstad et al. [14] through the use of lidar, Landsat, and soils information as input data, while other studies modeling black ash have relied on Landsat and radar [32] and the MODIS Terra data product [13]. Our findings support the use of GEE as a useful tool to integrate remote sensing datasets of different resolutions. As indicated in Tables 3 and 4, a longer time series was associated with higher relative accuracy compared to shorter time series. In particular, the use of polygon-based forest inventory data (i.e., the FIM data) provided excellent training data to detect ash presence across contrasting landscapes in Minnesota's diverse forest communities.

The spatial resolution of the classification map provides unique insights to the connectivity of forest cover types that is useful for targeting areas for strategic forest management. Geospatial information is ubiquitous in land management and landscape planning. The benefit of continuous imagery across boundaries circumvents some of the limitations of field-based measurements but does not replace the need for ground truth observations. This work provides a baseline of current ash abundance to attempt to quantify the risk of EAB infestation. The use of time series that date back to 1974 offers not only a more rigorous dataset, but also a window into where ash forests previously existed. It is possible that classification errors occurred due to the fact that older ash stands observed in historic time series have been succeeded by new cover types or transitioned to different species or a new ecosystem state associated with ash dieback [34]. These classification errors may be more prominent within the limited regions of where EAB has been observed in Minnesota since 2009 (i.e., in central and southern Minnesota).

The complexity of species composition in Minnesota forests make remote sensing techniques difficult, indicating that additional studies may be required to improve the accuracy of satellite time series analyses. More accurate models may be generated in these additional studies with increases in temporal and spatial resolution. The limitations of 30-m pixels are known to underrepresent highly mixed forests [35], and this would include ash or forests with a high understory density of ash trees. However, moderate resolution (i.e., 30 m) is likely sufficient to identify critical hotspots or connectivity analysis. The RandomForest classification probability was a highly skewed positive distribution where the majority of probability estimates were less than or equal to 0.1. Both the ash presence and absence error were minimized when 0.1 probability of ash classification was used as the cutoff threshold for ash presence. However, it is important to note that the relative abundance of ash on a plot will impact the detection of ash. Lower abundance of ash is more difficult to detect and, in these instances, the prevalence of misidentification of ash would increase.

Further research is required to understand how ash forest connectivity influences EAB dispersal capabilities. In particular, lidar-derived metrics such as the Compound Topographic Index are essential in determining dispersal capability as reflected in the presence of susceptible host trees. A wall-to-wall map of forest cover types and species distribution can guide selection of these target areas for further

investigation or management. In addition, and in particular with EAB, the time of insect arrival (if known) should also be considered because decreases in ash abundance generally begins six to seven years after EAB is first detected [7].

The detection of landscape change using satellite image time series is an instrumental tool for understanding forests. The access to free data such as Landsat and Sentinel and open research platforms such as GEE facilitates exponential growth in the realms of both research and management. Invasive species, impacts from climate change including shifting disturbance regimes, and the incorporation of multiple goals and objectives in forest management requires data on the composition and structure of forests. The continued evolution and merging of remotely sensed data with on-the-ground forest inventory data can aid managers in developing landscape-level plans. Whereas forest management often happens at a stand scale, the disturbance event (in this example, EAB), will impact the landscape and will cross ownership boundaries. County-, state-, and landscape-level maps promote communication across forest ownership boundaries. These mapping efforts facilitate the development of management plans that increase the health and resilience of forests.

## 5. Conclusions

In this study, the presence and abundance of ash species was predicted from a 30-m resolution satellite image time series combined with lidar-derived layers. The specific physiographic characteristics known to be suitable for ash trees were leveraged to aid classification. The annual phenologic fluctuation due to leaf drop and relative topographic position were key inputs to the RandomForest classifier. The highest success was in the detection of homogeneous black ash stands (72%) compared to the overall ash detection rate of 64%, which included three ash species. Time series analyses with satellite imagery provided a unique perspective on landscape characteristics that has promise for forest cover type and tree species classification.

Ash species make up a large portion of Minnesota's forests and are anticipated to be severely impacted by EAB in the coming years. The impact of losing ash trees will unquestionably alter Minnesota's landscape. Understanding the location and abundance of ash in Minnesota's forests is a requirement for ecologic preservation and fostering a future with healthy and resilient forests. This research represents our continual improvement in understanding the status and extent of ash in Minnesota that will lead to many more questions in the future.

**Author Contributions:** Conceptualization, M.B.R., M.A.W. and R.A.S.; methodology, T.K.H. and M.B.R.; software, T.K.H. and J.F.K.; validation, T.K.H. and R.A.S.; formal analysis, T.K.H.; investigation, T.K.H.; resources, T.K.H. and J.F.K.; data curation, T.K.H.; writing—original draft preparation, T.K.H.; writing—review and editing, M.B.R., M.A.W.-C., R.A.S. and J.F.K.; visualization, T.K.H.; supervision, M.B.R. and J.F.K.; project administration, M.B.R.; funding acquisition, M.B.R., M.A.W.-C. and R.A.S. All authors have read and agreed to the published version of the manuscript.

**Funding:** This research was funded by Minnesota Agricultural Experiment Station - Rapid Agricultural Response Fund and USDA Forest Service - Northern Research Station.

**Acknowledgments:** We thank Jennifer Corcoran of the Minnesota Department of Natural Resources for guidance with MNDNR inventory data. We thank Grant Domke and Richard McCullough for assistance with acquiring and interpreting Forest Inventory and Analysis plot data and Ram Deo and Christopher Edgar for assistance in interpreting state-level ash abundance. We thank three anonymous reviewers for their comments that help improve the quality of this manuscript.

**Conflicts of Interest:** The authors declare no conflict of interest. The funders had no role in the design of the study; in the collection, analyses, or interpretation of data; in the writing of the manuscript, or in the decision to publish the results.

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
