# Peer review of "Ash Presence and Abundance Derived from Composite Landsat and Sentinel-2 Time Series and Lidar Surface Models in Minnesota, USA"

_remotesensing, doi:10.3390/rs12081341_

Round 1

Reviewer 1 Report

General feedback

The manuscript entitled "Ash Presence and Abundance derived from Composite Landsat Time Series and Lidar Surface Models in Minnesota, USA" aimed to produce a 30-meter resolution map of current ash presence/absence in Minnesota and model abundance of ash in terms of a basal area using  Landsat imagery and lidar height metrics. 

In my opinion, this research is relevant and is in the scope of the journal, however, the manuscript must be improved.

Here are major comments:

  1. The introduction section must be extended. Besides the description of the current situation with Ash species and the effect of pest EAB on trees' morality, it is also needed to reflect in the Introduction previous experience of various Remote sensing applications for Minessota forest mapping as well as used classification algorithms. The importance and advantage of imagery time series in such studies should also be pointed out
  2. Methods section needs many improvements. In subsection 2.1.2. needs to be better explained how did you deal with Landsat 7 imagery that has gaps because of damaged sensor since 2003.  It also needs more details, how NDVI was used to identify forested areas (e.g. threshold value etc., line 106-107). How about Sentinel-2 data, have you done any corrections. As I know GEE platform can provide only top of the atmosphere reflectances. Because of different spatial resolution compared to Landsat images, how did you fix those issues?
  • Subsection 2.1.3. should also be extended. A better description of GEE is needed especially in case of work with big amounts of data, "User memory limit exceeded error" may occur. 
  • In subsection 2.2.1. a Random Forest classifier and its basic parameters/settings should be explained more in detail. 
  • In lines 147-148 it is said that only stands measured between 2000 and 2018 were included in this analysis to train RF classifier, how about Landsat and Sentinel imagery, were they selected during the same time period for each plot?

     3. In the Results section, when comparing the data in table 3 and 4, it is strange that Sentinel-2 shows the highest OOB accuracy (Tab. 3), but the lowest time-series NDVI accuracy (Tab. 4). This should be better explained.

4. In the Discussion section, would be nice to see a better analysis of findings, review similar studies and their accuracies. A more clear analysis of uncertainties and limitations description and could strengthen this paper.

Author Response

In response to comments from Reviewer 1:

NOTE: Our responses to review comments refer to line numbers in the Track Changes version of the submitted manuscript.

The manuscript entitled "Ash Presence and Abundance derived from Composite Landsat Time Series and Lidar Surface Models in Minnesota, USA" aimed to produce a 30-meter resolution map of current ash presence/absence in Minnesota and model abundance of ash in terms of a basal area using Landsat imagery and lidar height metrics.

In my opinion, this research is relevant and is in the scope of the journal, however, the manuscript must be improved.

We thank the Reviewer for their time and effort in reviewing the manuscript and providing comments. 

Here are major comments:

The introduction section must be extended. Besides the description of the current situation with Ash species and the effect of pest EAB on trees' morality, it is also needed to reflect in the Introduction previous experience of various Remote sensing applications for Minessota forest mapping as well as used classification algorithms. The importance and advantage of imagery time series in such studies should also be pointed out

L67-69: We added discussion of a recent publication that used the Landsat time series in Minnesota to map forest disturbances, and compare their approach to the challenges in mapping an individual species (as in our approach).

Methods section needs many improvements. In subsection 2.1.2. needs to be better explained how did you deal with Landsat 7 imagery that has gaps because of damaged sensor since 2003.  It also needs more details, how NDVI was used to identify forested areas (e.g. threshold value etc., line 106-107). How about Sentinel-2 data, have you done any corrections. As I know GEE platform can provide only top of the atmosphere reflectances. Because of different spatial resolution compared to Landsat images, how did you fix those issues?

L107-109: See here where we discuss gaps in Landsat 7 imagery.

L114-115: See here where we discuss identifying forested areas.

L120-124: We discuss clouds and poor-quality pixel corrections made in the analysis and how we resampled Sentinel-2 imagery to 30 m resolution.

Subsection 2.1.3. should also be extended. A better description of GEE is needed especially in case of work with big amounts of data, "User memory limit exceeded error" may occur.

L120-124: See here for our description of three ways we employed the Google Earth Engine platform. For us, the extent of the state of Minnesota was below the user memory limit at 30 meter resolution. 

In subsection 2.2.1. a Random Forest classifier and its basic parameters/settings should be explained more in detail.

L153-160: See here where we added information on the RandomForest approach. 

In lines 147-148 it is said that only stands measured between 2000 and 2018 were included in this analysis to train RF classifier, how about Landsat and Sentinel imagery, were they selected during the same time period for each plot?

The imagery time series variables of NDVI amplitude and NDVI phase were created using the length of the time series. They were not filtered by date.

  1. In the Results section, when comparing the data in table 3 and 4, it is strange that Sentinel-2 shows the highest OOB accuracy (Tab. 3), but the lowest time-series NDVI accuracy (Tab. 4). This should be better explained.

When comparing the accuracy between the four sensors, the relative error shows a similar pattern between OOB accuracy and detection accuracy. This pattern also aligns with the length of the time series, longer time series were associated with higher relative accuracy than shorter time series. We explain this on L270-272.

  1. In the Discussion section, would be nice to see a better analysis of findings, review similar studies and their accuracies. A more clear analysis of uncertainties and limitations description and could strengthen this paper.

L337-342: We added two sentences in the Discussion that compare our accuracy detection for black ash (72%) with that of two other studies from smaller geographic areas in Minnesota (85.5% and 89.5%). 

Reviewer 2 Report

Dear Authors,

Thank you for giving me a chance to review this article. It is timely and relevant and should be of interest to readers. It is timely because forest communities face a number of disturbance agents that occur slowly, may not be immediately detectable, and persist for years. Using time series analysis to address such scenarios is an increasingly useful tool for landscape assessment.

I liked that your article was relatively straight forward and to the point. However, in that brevity, I felt like a number of points needed some clarification. Also, often times an excessive number of figures and tables are presented, yet in this case some additional figure may have been useful. Additionally, there are some formatting and cartographic elements that could be addressed as well. Finally, there are also questions about training data and classification probabilities that may warrant more discussion. Overall, I thank you for your contribution and encourage you to refine this article.

Some of the main things that I felt needed clarification were related to the classification method, classification probabilities, training data selection and ratio of presence and absence representation, and validation data and techniques.

In terms of formatting, the last line of each page was cut off in the document I reviewed. This suggest the margins need to be adjusted but I cannot be certain this was not a printing error on my part. The map figures are presented with a title, and this should be reserved for the figure caption. In other words, I suggest you remove the title from the maps. You should also consider your legends. In some cases, elements are presented in the map that are not given in the legend, and in another case the range of values shown in the legend do not seem to match the values presented in the text. I may have misunderstood this, but I suggest you review and adjust.

Below are some specific comments and suggestions:

Line 126: As you begin to describe the point cloud data you refer to 1.5 pulss/m2. Starting here and later in the article the “m2” notation is used. I suggest addressing this and using the more appropriate form of m2, as is done in the

Line 128: When describing lidar derivatives the 3-m notation is used. Then 3 m. Please be consistent, and consider simply saying 3 meter, for example.

Line 139: The figure presented below this line does not need both a title in the map and a title in the figure caption. It is traditional to present the tile in the figure caption. Also, the caption describes a polygon database, yet there is no indication or illustration of polygons in the map. Please consider saying something like “Illustration of presence and absence of ash stands, based on the Minnesota Forest Inventory Management polygon database”. An inset illustrating the context of MN relative to the USA or North America may also be helpful. Also, why was location 1 chosen as an enlarged element? I understand the point of showing more detail, but is there some significance to this area?

Model Development:

Training data. It is stated that the ash training data came from a polygon database. Throughout this article, I had to wonder if the modeling unit an object or a pixel. When deriving training sites, were centroids of the polygon used, or were zonal statistics computed for the polygons? Were the selected sites reviewed to ensure that 1) if it was the centroid, that it fell in a location that resembled the phenomenon being mapped, or 2) that the polygon boundaries captured the phenomenon being mapped, and were not altered by disturbance or had unnecessary inclusions?

You describe a process for how ash sites were extracted from the polygon database, but I did not see reference to how non-ash sites were selected/extracted.

What ratio of presence versus absence sites did the final training data result in? It would certainly be good to clarify this, and it may be nice to repent that in a table. Later you state that classification probability was strongly right skewed. When using the RandomForest classifier, those distributions are often related to the proportion of records in the training classes. Please consider this and provide more detail in this section of the article.

In terms of image and topographic data. How did you handle the multiple resolutions of input data? This was not clear to me. In the table, on line 156 image data are all presented with 30 meter resolution. Sentinel-2 has 10 meter resolution of the bands in question. Were these data resampled before input to modeling. How? Was the 3 meter topographic data resampled to 30 meter resolution? Were all data used in their native resolution and internally handled to yield 30 meter output. Please consider giving more pre-processing details in this section. It will be useful for those considering the modeling process.

Line 172: The period near the end of the line seems out of place. Please consider removing it.

Line 174,176: Please address the m2 notation.

Line 178: Again, please format the map and remove the title. It does not belong there, and it is not sufficiently descriptive. Also, please consider adding a space between the table and the figure.

In the figure 3 caption, the CFI plots are said to be structure in stratified systematic fashion, but this is not described in the text above table 2. This should be addressed. Please give adequately describe the sampling protocol for each dataset.

Line 187: It is stated that coordinates were not used in the model. Such input can be useful when mapped features have some geographic gradient associated with them. It would seem important when trying to estimate geographic extent of a phenomenon. In this case, most of the sites in question are in close proximity, so it is understandable why this choice was made. So just commentary here.

Line 191: Concurrent with ash prediction map. Could you clarify a little here? Were basal area estimates only applied where ash was predicted to be present?

Line 193: It is fairly common, and rarely satisfying to use FIA data for mapping and validation. Throughout this section it was unclear whether sub-plots were used, or if the sub-plots were summarized to the central plot location. Were there 5,924 sites, or 5,924 plots summarized for a smaller number of sites? Were the relationship between the plot attributes and site location evaluated? The ground condition of the central plot location of an FIA plot can be very different than the summarized value of all sub-plots. This is am important component, so please consider describing this in more detail.

Line 216: It is stated that NDVI(a) and NDVI(m) were the most important classification variables. It is not clear how that is illustrated in table 3. Also, while out-of-bag is a term that those familiar with RandomForest will be comfortable with, it may be worth explaining how it is used in the text.

Overall, in this section it was not clear how the 10% classification probability threshold was selected. It may be useful to present a figure of the classification probabilities to help readers understand how this decision was made. Was it quantitative or professional judgement? In terms of classification probabilities, were these simply what is output by RandomForest? That is, are these the vote-based statistics associated with the binary classification?

Line 232: Again, remove the title form the figure itself. Within the figure, county boundaries are presented but not described by the legend. Either remove the county boundaries from the figure or provide a reference in the legend. Beyond that, one must ask what is the significance of the counties to this figure or even this article? There is no summary of ash presence by county, or even discussion of what counties have high or low abundance? Perhaps this is something to include in the article, especially since the topic is oriented towards management of an infestation in this particular state. If not, consider leaving counties out of it altogether.

Line 241: It is stated that the “overall abundance of ash basal area was 16.1 square meters per hectare on average”. This could likely be stated more clearly. Nonetheless, this is an important quantity when considering figure 5.

Figure 5: Again, please address the cartographic elements previous discussed. Remove the non-descriptive title, provide legend entry for counties or remove them since their significance is not discussed, but most importantly make the figure caption more informative, and consider the basal area scale presented in the legend. Pervious it was stated that the overall abundance was centered around 16.1 units, but the legend ranges from 20-55. Please address this.

Line 281: The sentence starting on this line could likely be written more clearly. Perhaps consider substituting “the application of” instead of “application for”.

Line 301: The notion of disturbance has been something I would have liked to be addressed throughout the article, especially in the context of time-series imagery. I was glad to see some acknowledgement of this concept and consequences of using contemporary training data for historic data.

Line 309: The skew of the probability distribution may be related to the ratio of presence and absence data. Please consider evaluating this.

Line 330: While the idea is appropriate, please consider rewording the sentence beginning half way through this line.

Good conclusion.

Author Response

In response to comments from Reviewer 2:

NOTE: Our responses to review comments refer to line numbers in the Track Changes version of the submitted manuscript. 

Thank you for giving me a chance to review this article. It is timely and relevant and should be of interest to readers. It is timely because forest communities face a number of disturbance agents that occur slowly, may not be immediately detectable, and persist for years. Using time series analysis to address such scenarios is an increasingly useful tool for landscape assessment.

I liked that your article was relatively straight forward and to the point. However, in that brevity, I felt like a number of points needed some clarification. Also, often times an excessive number of figures and tables are presented, yet in this case some additional figure may have been useful. Additionally, there are some formatting and cartographic elements that could be addressed as well. Finally, there are also questions about training data and classification probabilities that may warrant more discussion. Overall, I thank you for your contribution and encourage you to refine this article.

 We thank the Reviewer for their time and effort in reviewing the manuscript and providing comments. 

Some of the main things that I felt needed clarification were related to the classification method, classification probabilities, training data selection and ratio of presence and absence representation, and validation data and techniques.

In terms of formatting, the last line of each page was cut off in the document I reviewed. This suggest the margins need to be adjusted but I cannot be certain this was not a printing error on my part. The map figures are presented with a title, and this should be reserved for the figure caption. In other words, I suggest you remove the title from the maps. You should also consider your legends. In some cases, elements are presented in the map that are not given in the legend, and in another case the range of values shown in the legend do not seem to match the values presented in the text. I may have misunderstood this, but I suggest you review and adjust.

The manuscript appears to print fine for the authors without any lines cut off. We have improved the figure captions with additional explanations and clarity, including an important oversight we made to the caption in Figure 2.

We have updated the figures and have removed the titles in Figures 1, 3, and 4.

Below are some specific comments and suggestions:

Line 126: As you begin to describe the point cloud data you refer to 1.5 pulss/m2. Starting here and later in the article the “m2” notation is used. I suggest addressing this and using the more appropriate form of m2, as is done in the

 We changed all references to square meters to m2, with the 2 as a superscript.

Line 128: When describing lidar derivatives the 3-m notation is used. Then 3 m. Please be consistent, and consider simply saying 3 meter, for example.

 We changed to “3 meter resolution” throughout the manuscript.

Line 139: The figure presented below this line does not need both a title in the map and a title in the figure caption. It is traditional to present the tile in the figure caption. Also, the caption describes a polygon database, yet there is no indication or illustration of polygons in the map. Please consider saying something like “Illustration of presence and absence of ash stands, based on the Minnesota Forest Inventory Management polygon database”. An inset illustrating the context of MN relative to the USA or North America may also be helpful. Also, why was location 1 chosen as an enlarged element? I understand the point of showing more detail, but is there some significance to this area?

L164-166: We improved the figure caption using the reviewer’s suggestion and added “Inset displays an example of diverse Minnesota forests that transition from non-ash to ash-dominated.” We also removed the map title within the figure and added the location of Minnesota within the United States.

Model Development:

 Training data. It is stated that the ash training data came from a polygon database. Throughout this article, I had to wonder if the modeling unit an object or a pixel. When deriving training sites, were centroids of the polygon used, or were zonal statistics computed for the polygons? Were the selected sites reviewed to ensure that 1) if it was the centroid, that it fell in a location that resembled the phenomenon being mapped, or 2) that the polygon boundaries captured the phenomenon being mapped, and were not altered by disturbance or had unnecessary inclusions?

L147-148: In our analysis, the modeling unit is a pixel. We added text here about how we resampled polygons to fit the input variables. While not widespread in the forest types in Minnesota, disturbance or other landscape changing factors may be a source of error in the model predictions. 

You describe a process for how ash sites were extracted from the polygon database, but I did not see reference to how non-ash sites were selected/extracted.

Non-ash sites were identified by polygons that had different tree species labeled as the dominant species for the stand. On L173-174 we describe how the main cover type and primary species in the stand were used to select ash stands.

What ratio of presence versus absence sites did the final training data result in? It would certainly be good to clarify this, and it may be nice to repent that in a table. Later you state that classification probability was strongly right skewed. When using the RandomForest classifier, those distributions are often related to the proportion of records in the training classes. Please consider this and provide more detail in this section of the article.

L180-183: We mention here that approximately 2% of FIM polygons were ash-dominant, and also mention the number of polygons (total number and number of those dominated by ash).

In terms of image and topographic data. How did you handle the multiple resolutions of input data? This was not clear to me. In the table, on line 156 image data are all presented with 30 meter resolution. Sentinel-2 has 10 meter resolution of the bands in question. Were these data resampled before input to modeling. How? Was the 3 meter topographic data resampled to 30 meter resolution? Were all data used in their native resolution and internally handled to yield 30 meter output. Please consider giving more pre-processing details in this section. It will be useful for those considering the modeling process.

Sentinel-2 imagery was processed at 10 meter resolution but resampled to 30 meter resolution using the mean value resampling when combined with Landsat imagery. The topographic variables were created at 3 meter resolution and resampled to 30 meter resolution using the mean value prior to modeling. This is mentioned on L147-148.

Line 172: The period near the end of the line seems out of place. Please consider removing it.

 We removed the period.

Line 174,176: Please address the m2 notation.

 We changed all references to square meters to m2, with the 2 as a superscript.

Line 178: Again, please format the map and remove the title. It does not belong there, and it is not sufficiently descriptive. Also, please consider adding a space between the table and the figure.

 We removed the title from the map. We also added a space between the table and figure.

In the figure 3 caption, the CFI plots are said to be structure in stratified systematic fashion, but this is not described in the text above table 2. This should be addressed. Please give adequately describe the sampling protocol for each dataset.

L224-230: For Figure 3 and the paragraph preceding it, we have moved some of the information previously contained in the figure caption to the main text, in addition to providing more background on the inventory plots. For the CFI inventory dataset, we erred by mentioning it was a stratified design and did not include the correct plot size. We have fixed the description of the design and plot size.

Line 187: It is stated that coordinates were not used in the model. Such input can be useful when mapped features have some geographic gradient associated with them. It would seem important when trying to estimate geographic extent of a phenomenon. In this case, most of the sites in question are in close proximity, so it is understandable why this choice was made. So just commentary here.

We thank the reviewer for this insight. 

Line 191: Concurrent with ash prediction map. Could you clarify a little here? Were basal area estimates only applied where ash was predicted to be present?

 L236-237: Yes, we added that if ash was present, we determined basal area estimates. 

Line 193: It is fairly common, and rarely satisfying to use FIA data for mapping and validation. Throughout this section it was unclear whether sub-plots were used, or if the sub-plots were summarized to the central plot location. Were there 5,924 sites, or 5,924 plots summarized for a smaller number of sites? Were the relationship between the plot attributes and site location evaluated? The ground condition of the central plot location of an FIA plot can be very different than the summarized value of all sub-plots. This is am important component, so please consider describing this in more detail.

There are 5,924 FIA sites that each have a center plot and three sub-plots. To account for potential differences between the center and sub-plots, we chose the window of 9 pixels around the central plot that encompass all four sub-plots. If ash was present (presence was defined as at least 10% of total plot BA), the site was considered ash presence for validation purposes. The window of 9 pixels around each center plot was extracted to compare the classification result to ash presence in the four sub-plots. The pixels are not perfectly aligned to the plots but the window of pixels accounts for issues with alignment and positional accuracy of the center plot. 

Line 216: It is stated that NDVI(a) and NDVI(m) were the most important classification variables. It is not clear how that is illustrated in table 3. Also, while out-of-bag is a term that those familiar with RandomForest will be comfortable with, it may be worth explaining how it is used in the text.

 L158-160: We provided a sentence on how the out-of-bag estimates are used in the RandomForest algorithm. 

Overall, in this section it was not clear how the 10% classification probability threshold was selected. It may be useful to present a figure of the classification probabilities to help readers understand how this decision was made. Was it quantitative or professional judgement? In terms of classification probabilities, were these simply what is output by RandomForest? That is, are these the vote-based statistics associated with the binary classification?

For our analysis, the errors of commission and omission of ash presence were minimized when classification probability was 0.10. Hence, this was the chosen classification probability. We clarify this on L206-208.

Line 232: Again, remove the title form the figure itself. Within the figure, county boundaries are presented but not described by the legend. Either remove the county boundaries from the figure or provide a reference in the legend. Beyond that, one must ask what is the significance of the counties to this figure or even this article? There is no summary of ash presence by county, or even discussion of what counties have high or low abundance? Perhaps this is something to include in the article, especially since the topic is oriented towards management of an infestation in this particular state. If not, consider leaving counties out of it altogether.

We have removed the title to Figure 5. On L280-281, we have elected to maintain the maps with county boundaries and indicate them in the figure captions. We added text in the Discussion on L397-400 that describes that county-level estimates help promote cross-agency collaborations. Several counties in Minnesota own and control significant acreages of ash forestlands and have their own county-level land management teams, the primary reason for us seeking to keep the county boundaries.

Line 241: It is stated that the “overall abundance of ash basal area was 16.1 square meters per hectare on average”. This could likely be stated more clearly. Nonetheless, this is an important quantity when considering figure 5.

L291-292: Changed to “...basal area of the species averaged…” 

Figure 5: Again, please address the cartographic elements previous discussed. Remove the non-descriptive title, provide legend entry for counties or remove them since their significance is not discussed, but most importantly make the figure caption more informative, and consider the basal area scale presented in the legend. Pervious it was stated that the overall abundance was centered around 16.1 units, but the legend ranges from 20-55. Please address this.

 We have removed the figure titles in Fig. 5 and 6 and clarified the county boundaries in the figure captions. The figure captions are also more descriptive. 

We erred in the legend on Figure 6. In our previous version we did not convert the units from basal area in square feet per acre to basal area in square meters per hectare. It is now corrected and aligns with the 16.1 square meters per hectare value reported in the text.

Line 281: The sentence starting on this line could likely be written more clearly. Perhaps consider substituting “the application of” instead of “application for”.

L335: Changed to “The application of...:” 

Line 301: The notion of disturbance has been something I would have liked to be addressed throughout the article, especially in the context of time-series imagery. I was glad to see some acknowledgement of this concept and consequences of using contemporary training data for historic data.

L333-335: We added a sentence with reference to Vogeler et al. 2020 and Cohen et al. 2016 that highlights the challenges (and possibilities) of  using Landsat time series as a predictor of insect disturbances.

Line 309: The skew of the probability distribution may be related to the ratio of presence and absence data. Please consider evaluating this.

The proportion of dominant ash forests on the landscape are reflected by the ratio of presence and absence data. We assume that the proportion of species represented in the polygon data more or less represents reality. 

Line 330: While the idea is appropriate, please consider rewording the sentence beginning half way through this line.

L397-400. We broke these sentences into two and reworded them. 

Good conclusion.

 We appreciate the compliment!

Reviewer 3 Report

Brief summary:

The article describes a possibility to perform a wall-to-wall mapping of Ash presence with satellite time series and Lidar surface models. The abundance is estimated by ash basal area within 30m² pixels. A combination of Satellite based time series classification with random forest and a multivariate regression model on Lidar based Canopy height models is used. As reference and validation different local and county wide inventory data are used.

Broad comments:

The article is well written and describes the general approach for the implemented workflow. I would suggest going more in detail in some steps and in the discussion: this relates to the used reference data, methodology and validation.

Referring to the title: Only Landsat Time Series are mentioned. However Sentinel 2-data are used as well. I think it should be mentioned in the title as well: eg: …… composite Landsat and Sentinel 2 Time Series…..

Specific comments:

Line 18: Expression- “further inform”, doesn’t seem to be adequate, use “to improve or to enhance

Line 18: Expression – “These factors are used collectively”, doesn’t seem correct and to general to me. Not all input data are factors. Use “these input data are combined

Line 24: Ash basal area - It would be interesting here to mention also the RSME in % done by a cross-validation.

Line 58 – 64: You describe thoroughly the different vegetation types of black ash. It would be interesting to this in the validation and discussion.

Line 83: you mention three different approaches [10, 12, 13] to map Ash presence in Minnesota. It would be interesting to discuss the difference of your approach to these others in the discussion.

Line 85/86: Expression: Suggestion: separate in two sentences. Was the aim to model the abundance or was the aim the map and the model was one step to do it? “The Ash abundance was modeled by terms of basal area using lidar height metrics.”

Chapter 2.1.2:

Was there any preprocessing done with Sentinel-2 Data? If yes which, if no, why not and difference should be included in the discussion.

Line 103 – 106: Please give references to the following products and procedures: L1T, LEDAPS, LaSRC, CFMASK.

One is: - Masek, J.G., E.F. Vermote, N. Saleous, R. Wolfe, F.G. Hall, F. Huemmrich, F. Gao, J. Kutler, and T.K. Lim. 2013. LEDAPS Calibration, Reflectance, Atmospheric Correction Preprocessing Code, Version 2. ORNL DAAC, Oak Ridge, Tennessee, USA. http://dx.doi.org/10.3334/ORNLDAAC/1146

Line 114 (2012) is not necessary

Line 114/115: Which imageries belong to one cloud-masked imagery archive? Please precise how you define one imagery archive, e.g. by sensor?

Line 129: TPI and CTI, pleas explain how they are calculated and if in case give a reference.

Line 132/133: Last sentence can be deleted. It is already described in the beginning of the chapter.

Line 146: “The FIM dataset was filtered in several ways” - Please explain more precisely which filtering you performed.

Line 148/149 und Figure1: How did you deal with different mixture ratio of ash as main cover type and primary species in the stand? Did you only use pure Ash stands or above a certain proportion?

Line 145: FIM dataset. Please give a reference to the description of the FIM dataset and some explanation. It is of interest to know on which Data the aerial interpretation has been performed (ground resolution and spectral channes…) Was it done on Orthophotos or stereoscopically ? What were the criteria for a stand?

Line 151ff: Which pixels have been included in the training data -pixels which are completely or partially covered by the training polygons? How did you integrate or aggregate the data with 3mresolution to the 30m predictor variables? This questions concerns also Table 1 – Did you implement also some spatial alignment of the rasters?

Line 162: Change “Sentinel” to “Sentinel 2”

Line 165: include also the repetition rate as argument.

Line 163: Random Forest: How many trees did you use? What was your m?

Line 172 – 176: Were all these plots pure ash? What was the minimum DBH for CFI and the Black ash and EAB research plots?

Line 188 ff: How did you include the 3m Pixels to the respective overlapping Plot circles? Did you use only completely included pixels or also partially overlapping pixels? What was the spatial accuracy of the reference data? How was the localization done - by GNSS or aerial interpretation, or other tools?

Table 2: There is a time difference between the lidar data (2008-2011) and the reference data (2014-2018). Did you encounter this time difference in your regression model?

Line 190: You fitted your model on 404m² plots. How did you predict to the 30m (900m²) pixels?, did you do any transformation or aggregation?

Line 201: “True coordinates of FIA” What was the spatial accuracy of these data? Was the localization done by GNSS or aerial interpretation, or other tools?

Line 204: How was the relative proportion of ash incorporated in the validation?

Line 217: Separate in two sentences:….The accuracy varied by sensor origin.

Table 3: Why didn’t you show the results of Landsat 7 and Composite?

Line 223: How did you compose the time series/sensors?

There is a different time difference between the FIA reference data and the respective Time series (eg Sentinel is younger and Landsat is older. This is a point to consider in the discussion.

Table 4: it would be helpful to add a column with the length of the time series.

Chapter 3.1.2 Ash abundance:

Could you please show the formula of your model?

Did you do a cross validation? Could you calculate the RSME also?

Line 238: Do you mean by source, the different reference data?

Line 241: Is this the mean abundance of all pixels classified as Ash presence? Please specify

Line 254: Please give a reference, explaining the sampling design of the plot.

Line 252-254: How did you define ash absence in your reference data? Please describe. What was the overall number of FIA plots, the number of plots with ash absence, ash presence and if applicable the in the validation not included plots.

Line 262/263: I would suggest putting it into two phrases, to distinguish between presence and abundance. “….provided a new possibility to assess ash presence. Combined with lidar data the abundance in terms of basal area per ha is estimated.”

Line 302: What do you mean by …limited regions –….. these specific limited regions?

Line 330: Double blank before State

Line 339: Remove “More broadly,” it’s not necessary.

Author Response

In response to comments from Reviewer 3:

NOTE: Our responses to review comments refer to line numbers in the Track Changes version of the submitted manuscript.

The article describes a possibility to perform a wall-to-wall mapping of Ash presence with satellite time series and Lidar surface models. The abundance is estimated by ash basal area within 30m² pixels. A combination of Satellite based time series classification with random forest and a multivariate regression model on Lidar based Canopy height models is used. As reference and validation different local and county wide inventory data are used.

 Broad comments:

 The article is well written and describes the general approach for the implemented workflow. I would suggest going more in detail in some steps and in the discussion: this relates to the used reference data, methodology and validation.

Referring to the title: Only Landsat Time Series are mentioned. However Sentinel 2-data are used as well. I think it should be mentioned in the title as well: eg: …… composite Landsat and Sentinel 2 Time Series…..

 We thank the Reviewer for their time and effort in reviewing the manuscript and providing comments. We have added Sentinel-2 to the title of the manuscript. 

Specific comments:

 Line 18: Expression- “further inform”, doesn’t seem to be adequate, use “to improve or to enhance

L18: Changed to “improve”.

Line 18: Expression – “These factors are used collectively”, doesn’t seem correct and to general to me. Not all input data are factors. Use “these input data are combined

 L19: Changed to “These input data were combined...”

Line 24: Ash basal area - It would be interesting here to mention also the RSME in % done by a cross-validation.

We did not perform a cross-validation in this analysis. Instead, we evaluated the models using the Forest Inventory and Analysis dataset, a dataset not used in model fitting. 

Line 58 – 64: You describe thoroughly the different vegetation types of black ash. It would be interesting to this in the validation and discussion.

 L337-344: See here where we add to the discussion about the detection accuracy of black ash and comparison with other studies. 

Line 83: you mention three different approaches [10, 12, 13] to map Ash presence in Minnesota. It would be interesting to discuss the difference of your approach to these others in the discussion.

 L349-352: We added text that discusses three other studies that have used Landsat, radar, and MODIS data products in modeling black ash.

Line 85/86: Expression: Suggestion: separate in two sentences. Was the aim to model the abundance or was the aim the map and the model was one step to do it? “The Ash abundance was modeled by terms of basal area using lidar height metrics.”

 L89-90: Separated into two sentences.

Chapter 2.1.2:

 Was there any preprocessing done with Sentinel-2 Data? If yes which, if no, why not and difference should be included in the discussion.

Cloud masking was included for Sentinel-2. Since Sentinel-2 imagery originates from the European Space Agency (ESA), the processing and image products differ slightly from the Landsat program. We describe this on L203-206.

Line 103 – 106: Please give references to the following products and procedures: L1T, LEDAPS, LaSRC, CFMASK.

 L104: Beginning at the start of this paragraph, we added citations for the L1T, LEDAPS, LaSRC, and CFMASK.  

Line 114 (2012) is not necessary

 Removed.

Line 114/115: Which imageries belong to one cloud-masked imagery archive? Please precise how you define one imagery archive, e.g. by sensor?

 Each filtered image collection contained all valid pixels from a specific sensor.

Line 129: TPI and CTI, pleas explain how they are calculated and if in case give a reference.

L142-149: We describe that the Topographic Position Index (TPI) is a calculation of the elevation difference between each cell in a DEM to the mean elevation of the neighborhood of surrounding cells. We added the Moore et al. 1991 citation to describe the Compound Topographic Index and also add more description of it.

Line 132/133: Last sentence can be deleted. It is already described in the beginning of the chapter.

Removed.

Line 146: “The FIM dataset was filtered in several ways” - Please explain more precisely which filtering you performed.

L172-174: The FIM dataset was filtered by year of measurement greater than 2000 and such that the dominant cover type and primary species in the stand were identical. We added these descriptions here.  

Line 148/149 und Figure1: How did you deal with different mixture ratio of ash as main cover type and primary species in the stand? Did you only use pure Ash stands or above a certain proportion?

Stands with mixed ratios were not included in the ash stand training data. The thematic classification of cover type represents the dominant cover type but does not break down types by proportion.

Line 145: FIM dataset. Please give a reference to the description of the FIM dataset and some explanation. It is of interest to know on which Data the aerial interpretation has been performed (ground resolution and spectral channes…) Was it done on Orthophotos or stereoscopically ? What were the criteria for a stand?

L162: We added a citation to the FIM dataset. This dataset features are digitized onscreen over standard 24k digital raster graphics (DRGs) and U.S. Geological Survey (USGS) digital orthophoto quadrangles (DOQs).

Line 151ff: Which pixels have been included in the training data -pixels which are completely or partially covered by the training polygons? How did you integrate or aggregate the data with 3mresolution to the 30m predictor variables? This questions concerns also Table 1 – Did you implement also some spatial alignment of the rasters?

The modeling unit is a pixel. Training polygons were sampled at a 30 m resolution effectively converting polygons to raster to match the resolution of the input raster variables. All rasters were produced using the same coordinate reference system. 

Line 162: Change “Sentinel” to “Sentinel 2”

Changed. 

Line 165: include also the repetition rate as argument.

 We are unclear about what the reviewer is referring to here. If this refers to the acquisition years, we have included that information in Table 4.

Line 163: Random Forest: How many trees did you use? What was your m?

L196-197: Added: The RandomForest model parameters were set to 10 decision trees with 2 input variables per split (m). 

Line 172 – 176: Were all these plots pure ash? What was the minimum DBH for CFI and the Black ash and EAB research plots?

L210-219: The plots contained ash as well as other non-ash species. We added information on minimum diameters for the datasets in the paragraph beginning at this line. 

Line 188 ff: How did you include the 3m Pixels to the respective overlapping Plot circles? Did you use only completely included pixels or also partially overlapping pixels? What was the spatial accuracy of the reference data? How was the localization done - by GNSS or aerial interpretation, or other tools?

Pixels were included if the centroid of the pixel was within the plot circle. Locations of the field plot centers were recorded with high accuracy GNSS receivers such as Trimble R2 or equivalent.

Table 2: There is a time difference between the lidar data (2008-2011) and the reference data (2014-2018). Did you encounter this time difference in your regression model?

L339-344: The difference in time between lidar acquisition and field data is a potential source of error. We’ve mentioned this in the Discussion.

Line 190: You fitted your model on 404m² plots. How did you predict to the 30m (900m²) pixels?, did you do any transformation or aggregation?

The model was fitted on a per unit area value of basal area so that area is accounted for when applying the model to 30 meter resolution.

Line 201: “True coordinates of FIA” What was the spatial accuracy of these data? Was the localization done by GNSS or aerial interpretation, or other tools?

Most FIA plots are located using recreational-grade GPS systems. Previous research has shown that these typically have accuracies of less than 3–7 m (Anderson et al. 2009; West. J. Appl. For. 24(3): 128–136).

Line 204: How was the relative proportion of ash incorporated in the validation?

We used the relative proportion of ash on plots to determine where live ash comprised at least 10% of the basal area that were used to evaluate the classification map, described on L206-208.

Line 217: Separate in two sentences:….The accuracy varied by sensor origin.

L65-266: Done.

Table 3: Why didn’t you show the results of Landsat 7 and Composite?

Missing Landsat7 data was added to the table. The composite is a result of mosaicking the classification results following the RandomForest algorithm, so no Out-of-Bag error was calculated.

Line 223: How did you compose the time series/sensors?

The composite was a result of mosaicking the classification prediction results.  

There is a different time difference between the FIA reference data and the respective Time series (eg Sentinel is younger and Landsat is older. This is a point to consider in the discussion.

 L339-344: We mention the time differences between Sentinel-2, Landsat, and field inventory measurements here in the Discussion.

Table 4: it would be helpful to add a column with the length of the time series.

Table 4: We added a column labeled “Acquisition years”

 Chapter 3.1.2 Ash abundance:

Could you please show the formula of your model?

L284: We added the linear model here.

Did you do a cross validation? Could you calculate the RSME also?

We did not perform a cross-validation. Instead, we evaluated the models using the Forest Inventory and Analysis dataset, a dataset not used in model fitting. 

Line 238: Do you mean by source, the different reference data?

 By this, we mean it was a change of source to the reference data.

Line 241: Is this the mean abundance of all pixels classified as Ash presence? Please specify

 L291-292. By this, we mean a mean abundance for all pixels that were classified as ash presence. We’ve clarified here. 

Line 254: Please give a reference, explaining the sampling design of the plot.

 L291-292: Additional details on sampling design of plots can be found here.

Line 252-254: How did you define ash absence in your reference data? Please describe. What was the overall number of FIA plots, the number of plots with ash absence, ash presence and if applicable the in the validation not included plots.

Ash absence was defined as forested plots that did not contain an ash tree. There were 5,922 plots total, 1,254 plots contained at least one live ash tree, 4,668 plots did not contain any live ash trees. We list these sample sizes in Table 5.

Line 262/263: I would suggest putting it into two phrases, to distinguish between presence and abundance. “….provided a new possibility to assess ash presence. Combined with lidar data the abundance in terms of basal area per ha is estimated.”

 L313-316: We split these sentences in two per the Reviewer’s comment.

Line 302: What do you mean by …limited regions –….. these specific limited regions?

L368-369. We clarified by adding “in central and southern Minnesota…”

Line 330: Double blank before State

 Fixed.

Line 339: Remove “More broadly,” it’s not necessary.

 Removed.

Round 2

Reviewer 2 Report

Dear Authors,

Thank you for submitting your revised manuscript. It is clear you did a fair amount of work to address previous comments, and this is good progress. Based on my review, some additional work may be necessary.

Not related to the quality of the manuscript was the mismatch between line numbers in your comments and the line numbers in the revised manuscript.

You do not give indication of whether training data were visually or otherwise inspected for validity. It is a lot of work and requires good interpretation skills, however blind used of training data is irresponsible. I doubt you will retroactively do this, so this is just commentary.

154-155 another component to the algorithm is specification of the proportion of withheld data in order to generate the oob. If you discuss the number or trees and splits, why not include the notion of withheld data?

What proportion of data were withheld (i.e. 0.66)? Furthermore, it might be good for readers to understand why 10 splits and not 100 or 1,000 were used. Was this to conserve computing resources, or was it because the separation between classes was strong? I assume the square root of predictor variables rule was used to arrive at 2 splits, but this is never mentioned. While these elements are rudimentary to those familiar with RandomForests it may be helpful to readers attempting to replicate the study.

160 The number 1 in the right pane is obscured. Consider making this darker, bolder, whatever but try to make it clear.

FIA data. Again, no inspection to determine if plot and image data correspond. I lament this because all too often datasets are used without inspection/verification and that adds to uncertainty.

Your desire to illustrate county boundaries is understandable. However, there are still no summary statistics associated with the county units provided in this analysis. Do you suggest these will be conducted by the county management teams? Maybe state that. Your comment in the discussion only suggests that county, state, and landscape level maps facilitate collaboration. This is a good point but without some additional information associated with these land units is difficult to see why they are important to the reader. Without going into excessive dialog about county level management their inclusion seems like noise, and a bit ethnocentric. Including a table of presence / absence, and basal area summaries associated with counties may be a way to make them relevant and add value to your paper by serving as a reference to the county practitioners you wish to support. This is up to your discretion and merely my suggestion.

208-209. the plots in figure 3.2 do not appear to be arranged systematically but that could be an illusion. So are they, in fact, systematic but do not appear so? Or is there a different sampling scheme at play?

The legend in Figure 6 has not been fixed. Despite the author’s comments the values represented are still the same as the original. For mean of 16.1 to exist there must be values above and below it. The legend currently ranges from 20 to 55. Unless something is being misunderstood, this is a major oversight.

  1. Why the use of “modelling”? This is not wrong but is inconsistent with previous uses of “modeling’.
  2. Most similar to that of? No reference to a person or study is given, just the citation[14]. It seems like a name should be stated.

Author Response

Dear Authors,

Thank you for submitting your revised manuscript. It is clear you did a fair amount of work to address previous comments, and this is good progress. Based on my review, some additional work may be necessary.

We thank the Reviewer for their comments. 

Not related to the quality of the manuscript was the mismatch between line numbers in your comments and the line numbers in the revised manuscript.

We think the manuscript version that we submitted was altered slightly by the editorial office after we last revised it. The line numbers presented here refer to the Track Changes version.

You do not give indication of whether training data were visually or otherwise inspected for validity. It is a lot of work and requires good interpretation skills, however blind used of training data is irresponsible. I doubt you will retroactively do this, so this is just commentary.

While minimally reflected in the documentation, there was significant effort to ensure the training data was appropriate. This included (1) visual inspections of the polygon species attributes in comparison with best available elevation and aerial imagery, (2) programmatic validation to identify irregularities in the training dataset, and (3) detailed discussions of data quality with DNR staff (the originators of the data). These efforts led to the pre-analysis filtering steps that were taken, including removing polygons recorded before the year 2000 and confirming the species attribute was correctly populated. This reduced the likelihood of erroneous input data and maintained an unbiased selection of dispersed training data. 

154-155 another component to the algorithm is specification of the proportion of withheld data in order to generate the oob. If you discuss the number or trees and splits, why not include the notion of withheld data?

In our analysis, 20% of the data were withheld to determine out-of-bag accuracy. This was added to L157. While this was useful in the initial assessment of the classification, we view the validation with the independent FIA data to be a more robust assessment of accuracy. 

What proportion of data were withheld (i.e. 0.66)? Furthermore, it might be good for readers to understand why 10 splits and not 100 or 1,000 were used. Was this to conserve computing resources, or was it because the separation between classes was strong? I assume the square root of predictor variables rule was used to arrive at 2 splits, but this is never mentioned. While these elements are rudimentary to those familiar with RandomForests it may be helpful to readers attempting to replicate the study.

Model parameters were empirically tested to determine the appropriate number of trees and splits that produced the greatest accuracy. The input variables per split was determined by the standard square root of predictor variables (added on L191-192).

160 The number 1 in the right pane is obscured. Consider making this darker, bolder, whatever but try to make it clear.

We updated Figure 1 with a bold number 1.

FIA data. Again, no inspection to determine if plot and image data correspond. I lament this because all too often datasets are used without inspection/verification and that adds to uncertainty.

A subset of the FIA plots were visually checked for agreement and alignment with the classification map prior to the quantitative accuracy assessment. We recognize that errors in the reference data could impact the accuracy assessment, however, there is support for maintaining data quality in the FIA program and a wide network of users to report data quality issues. We worked with U.S. Forest Service personnel to appropriately evaluate the FIA data. 

Your desire to illustrate county boundaries is understandable. However, there are still no summary statistics associated with the county units provided in this analysis. Do you suggest these will be conducted by the county management teams? Maybe state that. Your comment in the discussion only suggests that county, state, and landscape level maps facilitate collaboration. This is a good point but without some additional information associated with these land units is difficult to see why they are important to the reader. Without going into excessive dialog about county level management their inclusion seems like noise, and a bit ethnocentric. Including a table of presence / absence, and basal area summaries associated with counties may be a way to make them relevant and add value to your paper by serving as a reference to the county practitioners you wish to support. This is up to your discretion and merely my suggestion.

We added the percentage of county ownership on L94-96 to provide some perspective and justification for including the county boundaries, i.e., that counties and other public land managers are the majority of Minnesota’s forestland ownership.  

208-209. the plots in figure 3.2 do not appear to be arranged systematically but that could be an illusion. So are they, in fact, systematic but do not appear so? Or is there a different sampling scheme at play?

Originally, in the 1950’s the Cloquet Forestry Center Inventory (CFI) inventory plots were systematically arranged. Since then, additional plots have been added and/or removed to the inventory as stand units have changed. We removed the word “systematic” from L212 to modify this.

The legend in Figure 6 has not been fixed. Despite the author’s comments the values represented are still the same as the original. For mean of 16.1 to exist there must be values above and below it. The legend currently ranges from 20 to 55. Unless something is being misunderstood, this is a major oversight.

This was the author's error in the last revision. (We didn’t replace it with the correct figure). A new Figure 6 with the correct legend ranging from 11 to 36 square meters per hectare now appears. 

312. Why the use of “modelling”? This is not wrong but is inconsistent with previous uses of “modeling’.

L316: Changed to “modeling”.

313. Most similar to that of? No reference to a person or study is given, just the citation[14]. It seems like a name should be stated.

L342: We changed to  “most similar to that of Engelstad et al. [14]...”